# Portable X-Ray Fluorescence as a Tool for Urban Soil Contamination Analysis: Accuracy, Precision, and Practicality

Eriell M. Jenkins[1], John Galbraith[2], Anna A. Paltseva[1,3*]

[1] Delta Urban Soils Laboratory, School of Geosciences, University of Louisiana at Lafayette, Lafayette, LA, 70504 USA
[2] School of Plant and Environmental Sciences, Virginia Tech, Blacksburg, VA, 24061 USA
[3] Departments of Agronomy and Horticulture and Landscape Architecture, Purdue University, West Lafayette, IN 47907, USA

*Correspondence to*: Anna Paltseva (apaltsev@purdue.edu)

**Abstract.** Urban agriculture has become an essential component of urban sustainability, but it often faces the challenge of soil contamination with heavy metal(loid)s like lead (Pb), arsenic (As), chromium (Cr), copper (Cu), manganese (Mn), nickel (Ni), and zinc (Zn). Traditional laboratory methods for detecting these contaminants, such as atomic absorption spectroscopy and other inductively coupled plasma techniques, are accurate but can be costly, time-consuming, and require extensive sample preparation. Portable X-ray fluorescence (PXRF) presents a promising alternative, offering rapid, in situ analysis with minimal sample preparation. The study reviews literature on PXRF analyzers to determine their accuracy and precision in analyzing heavy metal(loid)s in urban soils, with the goal of optimizing sampling, reducing laboratory costs and time, and identifying priority metal contamination hotspots. A literature review was conducted using Web of Science and Google Scholar, focusing on studies that validated PXRF measurements with alternate laboratory methods or certified reference materials (CRMs). This study reviews 84 publications to evaluate the accuracy and precision of PXRF in analyzing heavy metal(loid)s in urban soils. The review covers instrument types, ation methods, testing conditions, and sample preparation techniques. Results show that, when properly calibrated, particularly with CRMs, PXRF can achieve reliable accuracy. *Ex situ* measurements tend to be more precise due to controlled conditions, although *in situ* measurements offer practical advantages in urban settings. Portable XRF emerges as a viable method for assessing urban soil contamination by balancing accuracy and practicality. Future research should focus on optimizing sample preparation and calibration to further enhance PXRF reliability in urban environments. Ultimately, strengthening PXRF methodologies and supporting extension efforts through improved, accessible soil-testing tools, facilitating healthier urban soils, safer urban food production, and enhanced community well-being.

## 1 Introduction

Urban agriculture has gained importance as a sustainable approach to enhancing food security, revitalizing green spaces, and improving community well-being in cities. Yet, urban soils often contain harmful levels of heavy metal(loid)s such as lead (Pb), arsenic (As), cadmium (Cd), chromium (Cr), copper (Cu), nickel (Ni), zinc (Zn), manganese (Mn)and mercury (Hg), due to past industrial activities, vehicle emissions, and other anthropogenic sources (Adimalla et al., 2020; Chaney et al., 1984; Chen et al., 2005; Cheng et al., 2015a; Hu et al., 2013; Kosheleva and Nikiforova, 2016; Mielke, 2016; Morel et al., 2015; Pouyat and Mcdonnell, 1991; Wei and Yang, 2010; Wilcke et al., 1998). These contaminants pose significant risks to human health, particularly in areas where urban agriculture is practiced (Dumat et al., 2019), making the survey of soil quality crucial.

Traditional laboratory analysis of heavy metal(loid)s in soil includes the use of atomic absorption spectroscopy (AAS) and inductively coupled plasma (ICP) technology techniques such as mass spectrometry (ICP-MS), optical emission spectroscopy (ICP-OES), and atomic emission spectrometry (ICP-AES) (Margui Grabulosa, 2006; Paya-Perez et al., 1993; Shefsky, 1997; U.S. EPA., 1996; U.S. EPA, 1998). These methods, however, can be costly and time-consuming, requiring extensive sample preparation. They also involve acid digestion, which can be hazardous, if handled inappropriately, and result in the generation

of acid waste, contributing to unsustainable practices. Additionally, these processes can completely alter and contaminate samples due to chemical reactions and the use of corrosive acids (Lee et al., 2016; U.S. EPA, 1996). Standard x-ray florescence instruments are also used to study heavy metal(loid)s, but they are stationary and therefore cannot be taken to the field (Guilherme et al., 2008).

Among these challenges, portable X-ray fluorescence (PXRF) has emerged as a rapid and cost-effective alternative for heavy

metal(loid) detection and quantification in soil (Madden et al., 2022; Ravansari et al., 2020). X-ray fluorescence uses X-rays, generated by a cathode tube (Bruker Corporation, 2023), to probe the internal atomic structure of elements. The X-ray beam interacts with the electrons in the atoms of the sample, causing the excitation of inner-shell electrons. Upon electron excitation, orbitals become vacant and because this is an unstable configuration, the vacant orbitals are filled by other electrons. This process, known as fluorescence, involves electrons from orbitals with higher energy moving down to occupy the vacant orbitals

with lower energy. The energy lost by an electron when transferring from a high-energy orbital to a lower-energy orbital is emitted in the form of electromagnetic radiation. The energy of the emitted light is equal to the energy difference of the orbitals, which is in the frequency range of x-rays and is unique for each element. This in turn can be harnessed for elemental identification where by measuring the energies emitted, the XRF instrument can pick out the elements present in the sample. To ascertain the quantity of each element, the instrument or associated software analyzes the proportions of individual energies

detected. In essence, XRF enables a detailed elemental analysis of materials through the study of X-ray interactions with atomic structures (Bruker Corporation, 2023).

Portable XRF offers real-time results, requires minimal sample preparation compared to methods like acid digestion. According to Pham et al. (2020), the instrument detects approximately 20-23 chemical elements of concern simultaneously. Some others are not detectable due to their concentration being smaller than the limit of detection of the devices (Pham et al.,

2020). Portable XRF applications include laboratory use as well as *in situ* analysis of metal(loid)s in soils and sediments, thin films, paints, coatings, oils and liquids, and hazardous waste. It is a non-destructive analytical technique allowing both qualitative and quantitative analyses of sample composition (Kalnicky and Singhvi, 2001). As this method of soil analysis gains popularity, the range of associated instruments available and methodology used grows as well. While several studies (Butler et al., 2012; Cheng et al., 2015; McLaren et al., 2012; Landes et al., 2019; Romzaykina et al., 2024; Zhu and Weindorf,

2009) have demonstrated the effectiveness of PXRF in measuring heavy metal(loid) contamination in urban soil, there are some limitations with the technology.

Accurate interpretation of PXRF data critically depends on rigorous calibration and an appropriate understanding of detection limits. Rousseau (2001) emphasizes that instrument calibration using Certified Reference Materials (CRMs), with a chemical composition ideally close to the sample's targeted composition, is crucial in minimizing analytical uncertainty. The author also

argues that manufacturer-provided lower detection limits (LLDs) can be overly optimistic and differentiates between the Instrumental Limit of Detection (ILD)—a statistically derived theoretical detection capability—and the practical Limit of

Determination of a Method (LDM), which incorporates real-world factors such as sample heterogeneity, preparation, and measurement reproducibility (Rousseau, 2001). This perspective is particularly important in urban soil analyses, where variability in sample composition and preparation can significantly influence the accuracy and precision of PXRF measurements.

Urban soils present unique analytical challenges distinct from agricultural or natural soils due to their high heterogeneity, complex land-use histories, and diverse contamination sources such as industrial emissions, vehicular traffic, and urban waste (Pouyat et al., 2010; Yesilonis et al., 2008). These characteristics significantly influence the accuracy and precision of analytical tools like PXRF. Given the increasing use of urban soils for agriculture, recreation, and residential development, PXRF's ability to rapidly and cost-effectively detect contamination hotspots is particularly advantageous (Ravansari, 2020). However, its performance can vary widely depending on sample preparation, soil characteristics, and instrument settings. This review addresses a key question: How accurate, precise, and practically applicable is PXRF in assessing metal contamination in urban soils, and under what conditions does its performance vary? To answer this, we synthesize findings from peer-reviewed literature to: (1) critically evaluate the accuracy and precision of PXRF for urban soil contamination assessment; (2) identify and assess methodological factors affecting PXRF performance in both in situ and ex situ contexts; and (3) provide practical recommendations for optimizing PXRF use in urban soil studies, with emphasis on spatial analysis and field deployment. These insights aim to improve sampling strategies, reduce laboratory costs, and support the identification of contamination hotspots to inform urban soil management and promote safe urban agriculture.

## 2 Overview of PXRF Instrumentation and Methodologies in Urban Soil Analysis

### 2.1 Selection of source material

For this study, Web of Science and Google Scholar were selected as the primary search engines due to their comprehensive coverage of scientific literature, particularly in the fields of soil science and environmental studies. These databases were purposefully searched for the following keywords and phrases in varying combinations: *(p)xrf, urban soil(s), trace element(s), heavy metal(loid), (p)XRF in situ vs. ex situ, trace metal(s), ICP comparison, and (p)xrf accuracy and precision.* The use of these platforms ensured a broad and diverse range of sources, capturing both well-established and emerging research relevant to the application of PXRF in urban soil analysis. When narrowing down the sources available, focus was directed to peer-reviewed studies where there was either an alternate lab method or CRMs used to validate the PXRF measurements. These articles were able to provide details concerning the correlation of the results in juxtaposition to more traditional laboratory methods, which helped to convey the information needed to thoroughly determine the true accuracy and precision of PXRF measurements in soil research applications. Articles that provided background information on the PXRF and heavy metal pollution were also used in this review.

We reviewed articles published from as early as 1990, which was the earliest relevant study available. During the literature search, particularly on Google Scholar, a number of retrieved articles did not meet the predefined inclusion criteria and thus were excluded from the review. In Fig. 1, the large decrease in initial search results compared to articles selected is clear. The search includes articles published up to March 2024. Lastly, repeated articles were disregarded. Ultimately, a total of 84 publications were used for the review. The literature review focused on journal publications instead of including book chapters to ensure a consistent analysis and to utilize sources with more rigorous peer review standards.

In the sections below, we first review the instruments used, calibration methods, testing locations, homogenization methods, testing containers, testing modes, and testing times used among the research reviewed. Further the accuracy and precision of the PXRF, and the ways sample preparation, calibration, and testing methods may have influenced the results are discussed. Finally, we conclude with advantages, limitations, and practical recommendations for PXRF analysis in urban soil settings.

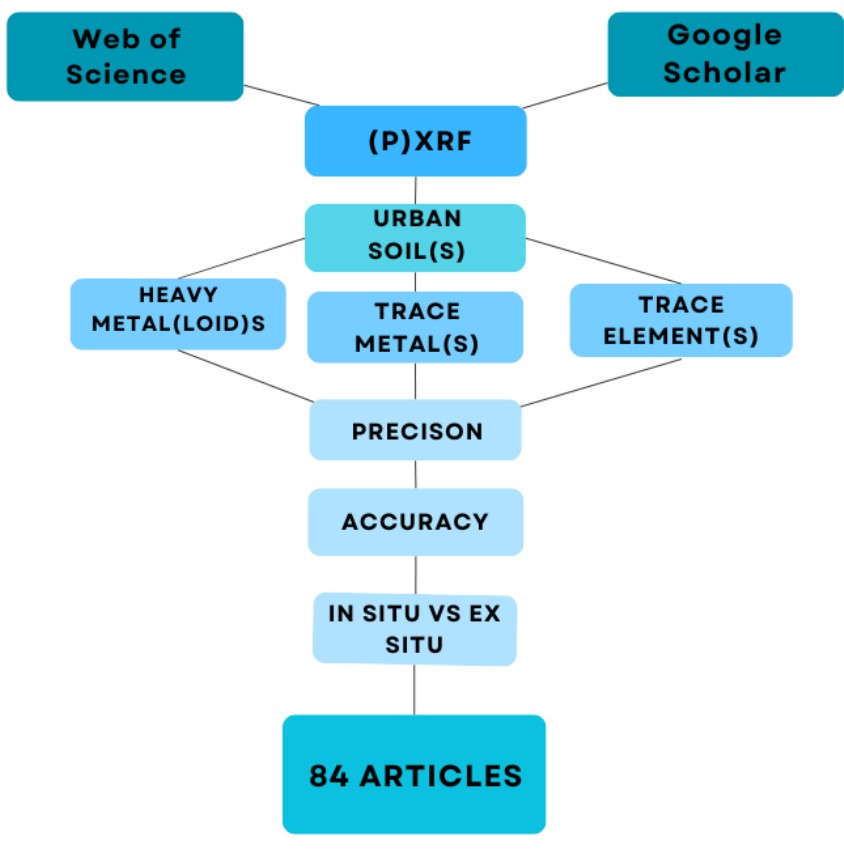

**Figure 1: A flowchart illustrating the keyword combinations used in the literature search for this review. Lines connecting terms indicate the use of the Boolean operator "AND," resulting in the selection of 84 articles.**

## 2.2 Instrument calibration

Calibrating PXRF instruments is a crucial step in accurately measuring heavy metal(loid) concentrations in soil samples, and various calibration methods have been explored in recent studies. Several studies have used Certified Reference Materials (CRMs) or soil standards to calibrate PXRF instruments. For example, in a study conducted by Qu et al. (2021) in Wuhan City, China, researchers verified the instrument by scanning the CRM GSS 3 seven times to calculate the relative standard deviation, which was found to be 6.51%. Another study by Schmidt et al. (2024) used three National Institute of Standards and Technology (NIST) reference materials: 2709 San Joaquin soil ($18.9 \pm 0.5$ mg/kg of Pb, $17.7 \pm 0.8$ mg/kg of As, $106 \pm 3$ mg/kg of Zn, and $34.6 \pm 0.7$ mg/kg of Cu), TILL-4p soil (50 mg/kg of Pb, 111 mg/kg of As, 70 mg/kg of Zn, and 237 mg/kg of Cu), and 2710 Montana Soil ($5532 \pm 80$ mg/kg of Pb, $626 \pm 38$ mg/kg As, $6952 \pm 91$ mg/kg of Zn, and $2950 \pm 130$ mg/kg of Cu). Romzaykina et al. (2024) calibrated their PXRF using the enclosed standard 2711A ($1400 \pm 10$ mg/kg of Pb, $107 \pm 5$ mg/kg of As, $414 \pm 11$ mg/kg of Zn, and $140 \pm 2$ mg/kg of Cu). Kim et al. (2019) utilized both a 316-alloy chip (Cr is 16.70%, Ni - 10.43%,Cu - 0.408%, and As - 0.0045%), blank samples, and NIST reference material 2710 Montana Soil.

Alloy chip, clip, or coin calibration were quite common among research available. This involves the use of small discs or chips made of the same alloy or metal(loid) as the sample being analyzed. The discs or chips are measured by the PXRF to establish a calibration curve for the specific alloy or metal(loid) (Pîrnău et al., 2020; Wang et al., 2015; Weindorf et al., 2016). This works similarly to the use of any other CRM. A study by Pîrnău et al. (2020) used alloy coins to calibrate a PXRF for the analysis of heavy metal(loid)s in soil samples from Romania. The PXRF was operated in laboratory based on manufacturer calibration and a factory pre-calibrated alloy coin (Alloy 316) was used to standardize the instrument before scanning.

Incorporating blanks and laboratory-specific reference standards into the calibration process is vital. Blanks—typically samples devoid of the targeted analytes—help detect background noise, cross-contamination, and instrument drift, ensuring measurement accuracy and reliability over time. Laboratory-prepared reference standards, closely matching local soil composition, enhance calibration accuracy by correcting for specific matrix effects unique to the samples being analyzed. Together, these calibration practices underpin the integrity and validity of both *ex situ* and *in situ* PXRF analyses across varied environmental settings.

## 2.3 Testing location

Soil analysis through PXRF has been widely used in research, with *ex situ* measurements being the most common approach. Although most studies have only utilized *ex situ* measurements (Kazimoto et al., 2018; Al Maliki et al., 2017; Mukhopadhyay et al., 2020; Suh et al., 2016), which involves taking soil samples to the laboratory for analysis measurements, some have

incorporated both field and laboratory PXRF measurements in their research (Lee et al., 2016; McStay et al., 2022; Urrutia-Goyes et al., 2017; Zhang et al., 2022). On the other hand, however, several studies consisted of measurements occurring strictly *in situ* (in the field) such as Jean-Soro et al. (2015), Paulette et al. (2015), Radu et al. (2013), Udeigwe et al. (2015), and Zhu and Weindorf (2009). The samples used in the studies we reviewed were taken from various locations such as residential areas, old and present mining sites, near roads, industrial areas, landfills, green houses, parks, and community gardens. In a study by Romzaykina et al. (2024), however, researchers calibrated portable XRF measurements using artificially contaminated soil mixtures representing diverse urban soils in Moscow, significantly improving accuracy to levels comparable to ICP-OES by applying tailored correction factors for Pb, Cu, Zn, and Ni; however, accuracy for Cd remained limited due to its high detection limit.

### 2.4 Homogenization

According to Schumacher et al. (1990), the need for sample homogeneity prior to laboratory analyses has been long recognized by geologists, chemists, and members of other scientific disciplines. Homogeneity is the degree that the material under investigation is mixed resulting in the random distribution of all particles in the sample. Scientists must strive to obtain a homogenous sample to obtain data exhibiting minimal error attributable to sample heterogeneity. It is especially important for elements such as Pb due to the inherent heterogeneity of soil Pb in a contaminated sample (Wharton et al., 2012). Among these, sieving and grinding are the most common techniques for achieving a uniform particle size distribution of soil samples. For instance, Williams et al. (2020) sieved the soil samples through a 2-mm sieve and then homogenized them using a ball mill. Similarly, Ravansari and Lemke (2018) homogenized the soil samples using a ball mill. Another commonly used homogenization method is grinding the soil samples using either a mortar and pestle or a mechanical grinder (Markey et al., 2008). Most authors used a combination of sieving through a 2-mm sieve and a grinding method to homogenize the soil samples. During laboratory tests, the mortar and pestle method was the most popular, with evidence of soil milling, mechanical grinding, being far less common. In some articles the homogenization method was not specified.

### 2.5 Testing container

When conducting PXRF measurements, various factors can influence the accuracy and precision of the results, including the type of container material used, as demonstrated in a study by Zambito et al. (2022) which found that the use of glass containers for soil samples resulted in elemental interferences compared to plastic containers. In one study, soil samples undergoing the XRF technique were analyzed through zip-locked plastic bags. The soil samples were measured according to an empty plastic bag analyzed as a blank sample and all sample measurements were blank-corrected (Wu, 2012). In another study, PXRF screenings were made in plastic Olympus cuvettes and covered with a "special" film (Romzaykina et al., 2024).

In a study by Laperche and Lemière (2020), several types of plastic films and plastic bags were tested for their effects on PXRF measurements. It was discovered that the use of certain types of plastic bags, such as polyethylene or polypropylene, can produce a significant interference signal that leads to inaccurate measurements. Taking measurements at different places

on a sample bag was proposed to control sample homogeneity. The authors investigated several types of plastic bags commonly used such as low-density polyethylene (LDPE) bags, a proprietary type of bag, and Prolene film on cups. Laperche and Lemière (2020) concluded a minor effect on heavier elements, and observable and major effects on elements like potassium, calcium,

and silicon. Their research further showed that plastic material could also contain some metal(loid)s. This suggests that LDPE is more suitable than high-density polyethylene, due to a lower number of additives. A variable number of layers of polyethylene film were tested against adsorption and reported a linear correlation with the number of layers. Ultimately, the authors recommended the use of low-density polyethylene bags as well as polypropylene testing films, as they produced the most accurate results with minimal interference.

Parsons et al. (2013) research suggests that external films and windows, such as Kapton and Mylar, that are used to protect the instrument may attenuate and scatter radiation, affecting analysis. For lighter elements like potassium and calcium, which emit lower energy radiation, attenuation effects by protective films can be significant due to the fact that the absorbance of low-energy X-rays is higher, which affects lighter elements more than heavier ones.

## 2.6 Testing mode

A "soil mode" is a specific testing mode in PXRF instruments that is designed for the analysis of soil samples. It uses a soil, sediment, and dust-specific calibration model and parameters (Lemière, 2018). This was the most popular mode used in the research analyzed in this review. According to Lemière (2018), the soil and mining modes were introduced in PXRF instruments to improve the accuracy of the analysis for different types of matrices. The soil mode is designed to provide analysis of light elements and to reduce the effect of heavy matrix elements on the analysis. In other words, a "soil mode"

offers broad and easy coverage of low concentrations and is often used for scanning and detection (Lemière, 2018).

In contrast, a "mining mode" is optimized for heavy elements and is suitable for the analysis of geological samples, such as ores, concentrates, rocks, soils, and other geological materials typically associated with mining and mineral exploration. Therefore, the best approach for high concentrations and quantification is user calibration with the "mining mode" (Lemière, 2018). It is characterized by a higher voltage and lower current than the "soil mode", which enables the instrument to detect

elements at higher concentrations (Goodale et al., 2012). The most popular factory testing mode used was a "soil mode", but in some articles, the mode was not specified. As a result, it was difficult to determine the effect of these modes on the results obtained from multiple studies. Thus, it would be beneficial for researchers to explore this further.

## 2.7 Testing Time

Studies have reported using analysis times ranging from 20 seconds to 5 minutes per sample. Jeong et al. (2021) used the

average readings of three 60 second PXRF readings. In a study conducted by Li et al. (2018), researchers analyzed 74 compost samples with measurement times of 180 seconds per sample but discovered the measurement time could be shortened to 90 seconds per sample by using a "soil mode" calibration rather than a "mining mode." Similarly, Xia et al. (2022) measured for

60 seconds per sample in a "soil mode." Liu et al. (2022) tested 88 samples for 60 seconds each (30 seconds per beam). Authors claimed that the longer measurement time was necessary due to the low concentrations of heavy metal(loid)s in their samples. Qu et al. (2021) tested the 93 samples used in their research for 90 seconds each. Another study of artificially spiked soils found that exposure times of 120 and 180 seconds yielded nearly identical readings for Cu, Pb, Ni, and Zn, because those contents were significantly above the detection limit (i.e., samples need to be measured for longer time if the concentrations are closer to the detection limits), while a 90-second exposure showed significant deviations, especially at lower concentrations (Romzaykina et al., 2024). Overall, the average time for PXRF testing was 90 seconds, most often followed by two repetitions per sample. When using a "soil mode", 90 seconds should provide a thorough reading for elements within a detectable limit.

Figures 2 and 3 illustrate how extending measurement time improves detection limits in PXRF experiments, based on Kalnicky and Singhvi (2001). Figure 2 show specific examples for heavy metals (Pb, Hg, Cd, and As), demonstrating how the detection limit, while changing differently for each metal, overall exhibit a strong reduction on the order of 80%. For example, a 15 sec measurement sets a detection limit of ~60 mg/kg for Pb but prolonging the measurements to 480 sec reduces the limit to ~10 mg/kg of Pb. It should be noted that the exact detection limits and measurement times depend on many factors, including instruments, soil porosity, and other external factors, however, the increase in sensitivity with increase in measurement time is a universal physical process that obeys Poisson's statistics. This occurs when a process has a constant rate of recurring events, such as absorption and reemission of x-ray, is recorded for a prolong period. This is demonstrated in Figure 3 that shows a relative detection limit, average for 10 metals, normalized to a baseline measurement of 15 seconds. The detection follows a Poisson distribution, where the relative limit decreases with the square root of the measurement time. Increasing the measurement time by a factor of 4 enables doubling of the sensitivity of the technique. Both figures underscore that increased measurement durations substantially lower detection limits and improve analytical precision, closely matching statistical predictions.

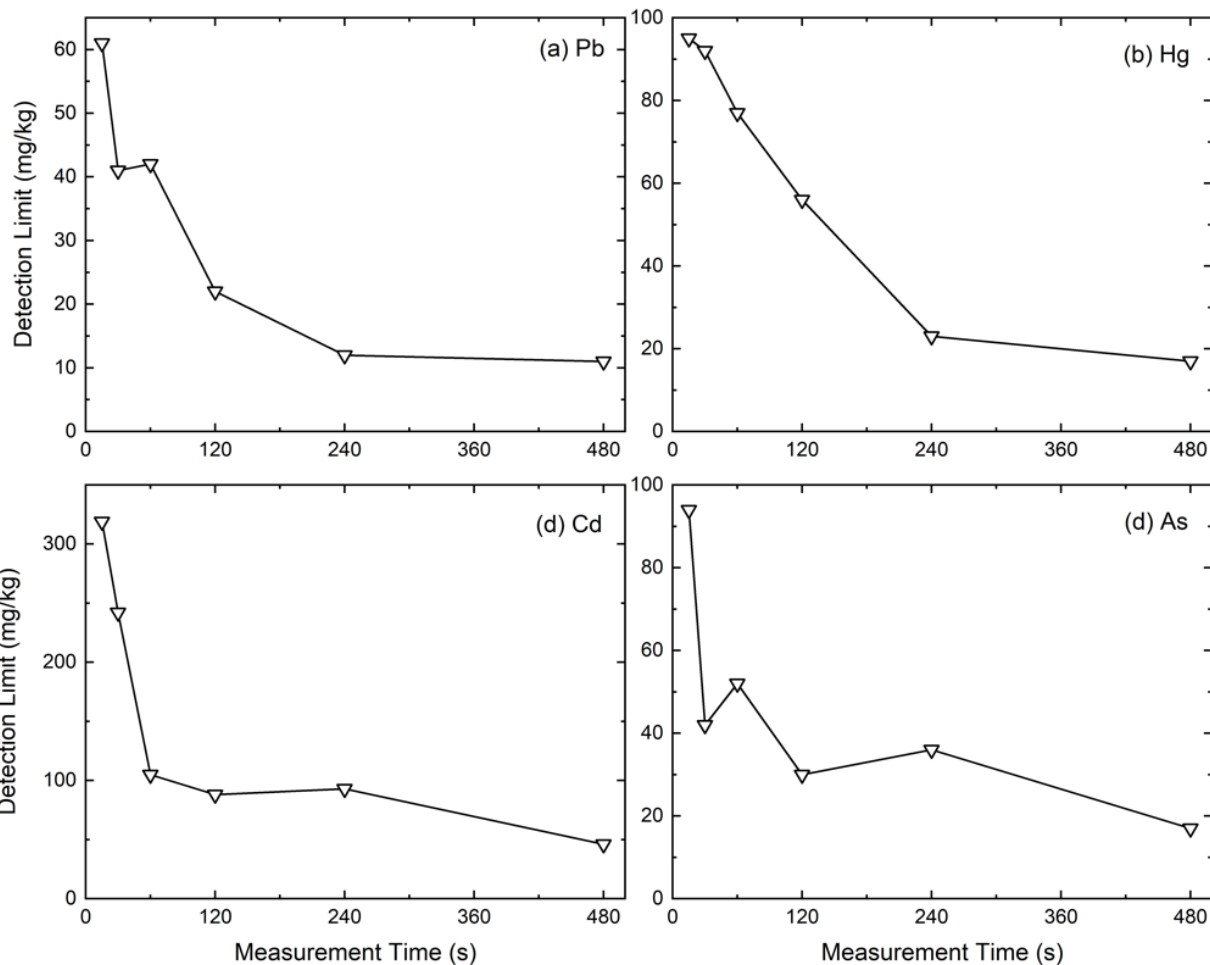

Figure 2: Effect of measurement time on detection limits (mg/kg) for Pb, Hg, Cd, and As, demonstrating significant decreases in detection limits with increased measurement durations, particularly noticeable within the initial 120 seconds (adapted from Kalnicky and Singhvi, 2001).

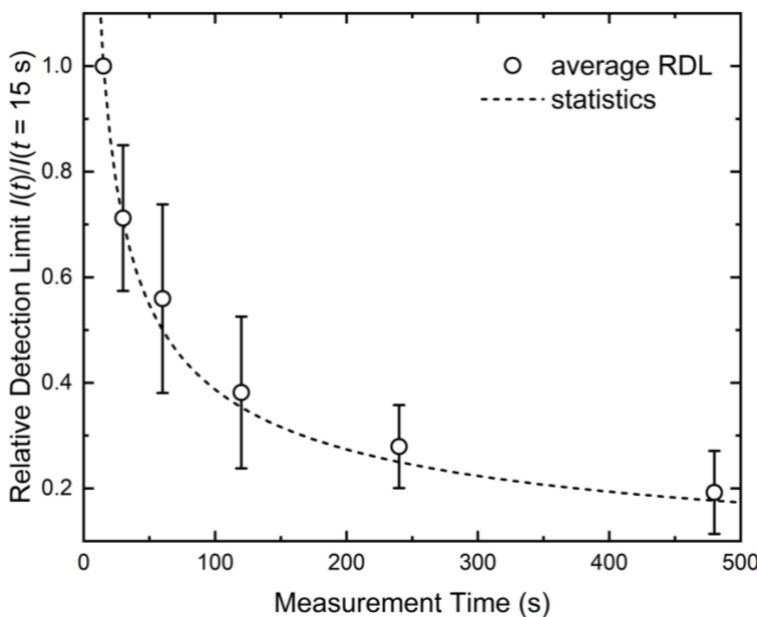

Figure 3: Relative detection limit, normalized to the limit with a measurement time of 15 s, averaged for 10 elements, Pb, Hg, Cd, As, Mn, Co, Ni, Cu, Zn, and Mo, taken from (Kalnicky and Singhvi, 2001). With increasing measurement time, the detection limit decreases following a Poission distribution with DL ~ 1/sqrt(t), as seen by the excellent agreement between the data and the fit, shown as a dashed line.

## 3 PXRF vs. Lab Analysis: A Comparative Perspective in Soil Testing

### 3.1 Comparison of PXRF and laboratory methods

Most of the studies that include *ex situ* measurements validate their results using AAS, ICP-MS, ICP-OES, and ICP-AES, as well as by comparing PXRF results to certified values. Several studies produced accurate and precise measurements. In the reviewed articles, the correlation coefficient (r) values between the PXRF results and laboratory analysis results for heavy metal(loid)s varied depending on the specific metal(loid). In a study conducted in Australia, McLaren et al. (2012) using Bruker Tracer III-V PXRF found that the PXRF measurements for As, Ca, Cr, Cu, Fe, K, Mg, Mn, Ni, P, Pb, Si, Ti, and Zn were highly correlated with the laboratory measurements, with correlations ranging from 0.82 to 0.98. Butler et al. (2007) found that PXRF measurements of Pb in soil samples had a correlation coefficient of 0.6 and 0.9 when compared to laboratory analysis using ICP-MS, indicating a strong positive relationship between the two methods. Cheng et al. (2015) found a positive correlation of 0.94 between two sets of Pb concentration data (ICP-MS and XRF Innov-X Delta Classic screening results on air-dried samples measured through Ziploc bags). These positive correlation coefficients display how PXRF results can obtain results comparable to more traditional forms of heavy metal analysis.

Gonzalez et al. (2021) conducted a study where they compared Pb measurements of a set of soil samples between PXRF (Niton XL2) and ICP-MS as well as between PXRF and two separate ICP-OES measurements, one following nitric acid extraction and the other following the relative bioaccessibility leaching procedure. They then calculated the relationship between the methods using the Berry-Mielke's Universal R test. An R value of 0.832 was the result of the PXRF vs. ICP-MS measurements. The R-value was 0.765 and 0.522 between the PXRF and both ICP-OES measurements, respectively. Note that the p-value for all aforementioned measurements was <0.0001. Therefore, ICP-MS was shown to have the strongest agreement with the tool.

Some studies found slightly weaker correlations, such as the study by McStay et al. (2022), which reported $r^2$ values ranging from 0.03 to 0.89 for Pb, Cu, and Zn, Mn, and As in urban soils using X-200 XRF (SciAps Inc). Alternatively, research by Gutiérrez-Ginés et al. (2013) on abandoned mines and landfills found that PXRF measurements had high $R^2$ values of As, Ca, Cd, Cr, Cu, Fe, K, Mn, Ni, Pb, Rb, Sr, Ti, V and Zn when compared to certified standards, ranging from 0.744 to 0.999. According to research by Kim et al. (2019), when comparing PXRF (Delta Premium PXRF spectroscope Olympus Innov-X Systems Inc., USA) measurements with ICP-AES analysis, PXRF tended to underestimate Pb and As concentrations, although a significant correlation between the two methods suggested that PXRF data can be used as a secondary variable despite its lower accuracy.

In a study conducted along a polluted river (Wu et al., 2012), researchers conducted pairwise comparisons between PXRF and ICP-AES measurements. The XRF used was NITON XL-722, which is equipped with a Cd-109 radioisotope source and Am-241 radioisotope source. The results suggest PXRF performs reasonably well for detecting metals like Pb and Zn in soils (typical urban contaminants), moderately for Cu and Ni, but is considerably weaker for accurately quantifying elements like As, Cr, Cd, and Hg. These differences likely arise from detection limits, soil matrix effects, or inherent limitations of the PXRF method. These findings were further supported by statistical analyses, including regression slopes and correlation coefficients, and are shown in Table 1. The importance of verifying and calibrating analytical methodologies was emphasized.

**Table 1: Adapted from Wu et al. (2012), summarizing soil contamination from runoff in Tainan City based on 60 samples collected and analyzed. Relative Proximity (%) refers to how well XRF identified samples exceeding the pollution threshold limit (PTL) compared to ICP-AES. It represents the percentage of samples detected above PTL by ICP-AES relative to those detected by XRF, indicating the effectiveness of XRF in accurately identifying contaminated samples requiring monitoring.**

| Element | Method | Mean (mg/kg) | $R^2$ | Relative Proximity (%) |
|---------|--------|--------------|-------|------------------------|
| **Pb** | ICP-AES | 2306.92 | 0.6689 | 85.17 |
| | XRF | 1371.6 | | |
| **Zn** | ICP-AES | 12184.91 | 0.6551 | 80.0 |
| | XRF | 24187.79 | | |

| | | | | | |
|---|---|---|---|---|---|
| **Ni** | ICP-AES | 444.05 | 0.7281 | 50.0 | |
| | XRF | 1590.92 | | | |
| **Cu** | ICP-AES | 3988.1 | 0.4095 | 35.42 | |
| | XRF | 4055.38 | | | |
| **As** | ICP-AES | 9.11 | 0.3449 | 25.0 | |
| | XRF | 66.13 | | | |
| **Cr** | ICP-AES | 78.38 | 0.1504 | 16.67 | |
| | XRF | 495.13 | | | |
| **Cd** | ICP-AES | 1.24 | 0.07823 | 5.77 | |
| | XRF | 31.43 | | | |
| **Hg** | ICP-AES | 0.7 | 0.01143 | 2.3 | |
| | XRF | 44.10 | | | |

Burlakovs et al. (2015) also compared data from PXRF (Olympus DELTA DS-2000) analysis with AAS and ICP-MS for 48 topsoil samples collected from the Kudjape Landfill in Estonia. Results showed a strong correlation between PXRF and AAS, as demonstrated by an $R^2$ value of 0.8915, indicating that PXRF provides reliable concentration estimates, especially for metals like Cu and Mn, which closely aligned with AAS measurements. However, some variability was noted for elements like Pb and Cr. While AAS and ICP-MS offer higher precision, PXRF was found to be satisfactory for screening purposes, offering significant time and resource savings in landfill management. For specific elements or higher precision, AAS or ICP-MS may still be necessary, and careful planning and sample preparation are essential for ensuring reliable PXRF results.

In Gutierrez-Gines et al. (2013) research, Zn, Pb, Ni, and As exhibit relatively stable PXRF/Standard ratios across their concentration ranges, with only minor fluctuations observed. In contrast, Cr and Cu show the largest variability, particularly at higher concentrations, indicating potential inconsistencies in PXRF measurements for these elements. Meanwhile, Cd and Mn demonstrate more stable ratios over smaller concentration ranges, with Cd showing a slightly increasing trend at higher concentrations. Figure 4, created based on Gutierrez-Gines et al. (2013) data, highlights these trends across varying concentration ranges, showing the relationship between PXRF measurements and certified values for each element.

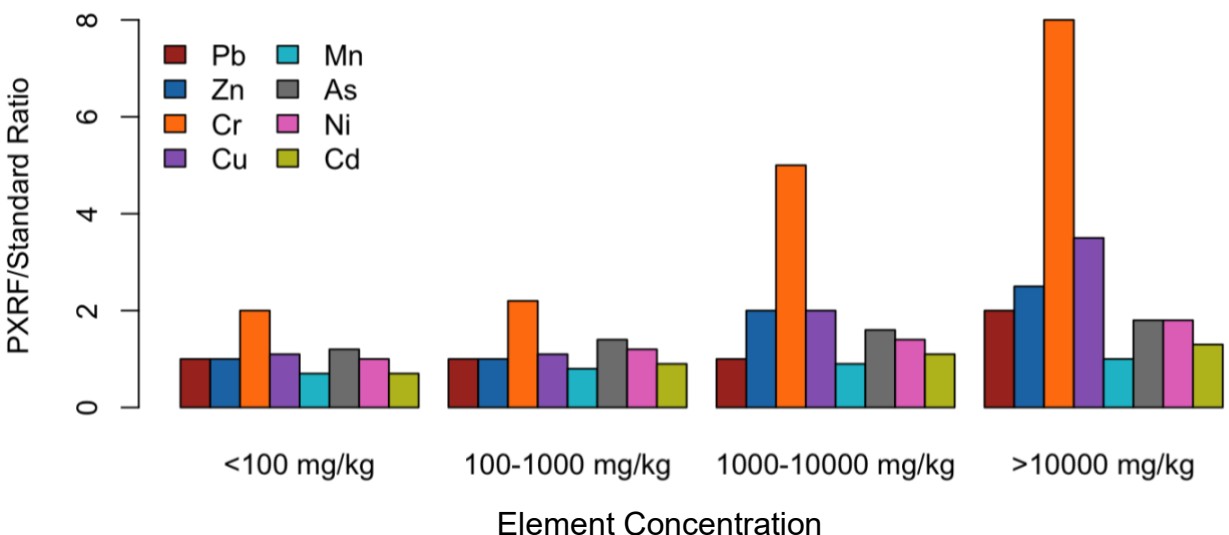

**Figure 4: Comparison of PXRF/Standard ratios for various elements (Pb, Zn, Cr, Cu, Mn, As, Ni, Cd) across different concentration ranges. This figure was created based on the data presented in Gutierrez-Gines et al. (2013), showing the relationship between PXRF measurements and certified values.**

The coefficient of variation for ICP-MS Pb measurements in Qu et al. (2021) study was lower at 30.07% compared to the one for *in situ* PXRF, which stood at 38.71%. Despite these variations, both methods yielded average Pb concentrations slightly below the background level of 35 mg kg⁻¹ in China. Interestingly, the average concentration of Pb measured by ICP-MS exceeded that of *in situ* PXRF, indicating the influence of other soil factors on PXRF analysis. This discrepancy underscores the necessity of correcting *in situ* PXRF measurements before utilizing them for spatial simulations of soil Pb contamination.

**Table 2: Summary of PXRF measurement correlations (R²) of key metallic elements with alternate laboratory methods, CRM's, and SRM's. The data in the table is based on various comparative studies used in this literature review.**

| Element | Comparison | Method | R² | Study |
|---|---|---|---|---|
| As | ICP-OES | Hydrochloric & Nitric Acid Digestion | 0.63 | McLaren et al., 2012 |
| As | ICP-MS | Not specified | 0.96 | Parsons et al., 2013 |
| As | ICP-MS | Not specified | 0.583 | Schmidt et al., 2024 |
| As | ICP-MS | Not specified | 0.84 | Schmidt et al., 2024 |
| As | CRM | USEPA Method 3051 | 0.96 | Ravansari et al., 2018 |
| Cr | ICP-OES | Hydrochloric & Nitric Acid Digestion | 0.41 | McLaren et al., 2012 |
| Cr | ICP-AES | Not specified | 0.82 | Schneider et al., 2015 |
| Cr | AAS | Nitric Acid & Hydrogen Peroxide Digestion | 0.89 | Burlakovs et al., 2015 |
| Cr | CRM | USEPA Method 3051 | 0.96 | Ravansari et al., 2018 |
| Cu | CRM | NIST 2711a | 0.9 | Li et al., 2018 |
| Cu | ICP-OES | Hydrochloric & Nitric Acid Digestion | 0.7 | McLaren et al., 2012 |
| Cu | ICP-AES | Not specified | 0.98 | Schneider et al., 2015 |
| Cu | ICP-AES | Not specified | 0.99 | Suh et al., 2016 |
| Cu | ICP-AES | 0.1N Hydrochloric Acid Digestion | 0.99 | Lee et al., 2016 |
| Cu | AAS | Nitric Acid & Hydrogen Peroxide Digestion | 0.89 | Burlakovs et al., 2015 |
| Cu | CRM | USEPA Method 3051 | 0.96 | Ravansari et al., 2018 |
| Hg | ICP-OES | EPA method 3050 B | 0.8 | McComb et al., 2014 |
| Mn | ICP-OES | Hydrochloric & Nitric Acid Digestion | 0.81 | McLaren et al., 2012 |
| Mn | ICP-AES | Not specified | 0.89 | Schneider et al., 2015 |
| Mn | AAS | Nitric Acid & Hydrogen Peroxide Digestion | 0.89 | Burlakovs et al., 2015 |
| Mn | CRM | USEPA Method 3051 | 0.96 | Ravansari et al., 2018 |
| Ni | ICP-OES | Hydrochloric & Nitric Acid Digestion | 0.78 | McLaren et al., 2012 |
| Ni | CRM | USEPA Method 3051 | 0.96 | Ravansari et al., 2018 |
| Pb | ICP-MS | USEPA Method 3051 | 0.89 | Al Maliki et al., 2017 |
| Pb | AAS | SW846-7420 and NIOSH 7082 | 0.98 | Markey et al., 2008 |
| Pb | ICP-MS | EPA method 1340 | 0.99 | Landes et al., 2019 |
| Pb | ICP-OES | Hydrochloric & Nitric Acid Digestion | 0.83 | McLaren et al., 2012 |
| Pb | ICP-AES | Not specified | 0.99 | Schneider et al., 2015 |
| Pb | ICP-MS | USEPA Method 3050B | 0.93 | Walser et al., 2022 |
| Pb | ICP-OES | USEPA Method 3050B | 0.81-0.94 | Zhang et al., 2022 |
| Pb | ICP-MS | Not specified | 0.973 | Schmidt et al., 2024 |
| Pb | AAS | Microwave-assisted strong acid extractions | 0.99 | Shefsky, 1997 |
| Pb | ICP-AES | Microwave-assisted strong acid extractions | 0.98 | Shefsky, 1997 |

| | | | | | |
|---|---|---|---|---|---|
| Pb | SRM | Microwave-assisted strong acid extractions | 1 | Shefsky, 1997 |
| Pb | ICP-AES | 0.1N Hydrochloric Acid Digestion | 0.99 | Lee et al., 2016 |
| Pb | AAS | Nitric Acid & Hydrogen Peroxide Digestion | 0.89 | Burlakovs et al., 2015 |
| Pb | CRM | USEPA Method 3051 | 0.96 | Ravansari et al., 2018 |
| Zn | CRM | NIST 2711a | 0.6 | Li et al., 2018 |
| Zn | ICP-OES | Hydrochloric & Nitric Acid Digestion | 0.89 | McLaren et al., 2012 |
| Zn | ICP-AES | Not specified | 0.96 | Schneider et al., 2015 |
| Zn | CRM | USEPA Method 3051 | 0.96 | Ravansari et al., 2018 |


A study conducted by Pozza et al. (2020) combined the use of PXRF and visible near-infrared technology. They first performed Cubist modelling, a statistical machine-learning method used for predicting continuous numeric outcomes from large datasets by creating predictive rule-based models, which helped them obtain predictions of the results. The resulting data exhibited high skewness, with the PXRF having higher values for Lin's Concordance correlation coefficient, which is a statistical

measure that quantifies agreement between two measurement methods, indicating both accuracy and precision simultaneously. However, the most accurate results were achieved by incorporating both visible near infrared principal components and PXRF Compton-normalized values through a generalized regression analysis model. Since model averaging of visible near infrared and PXRF predictions outperformed individual methods, integrating the two improved Pb prediction accuracy in this case (Pozza, et al., 2020).


The results obtained by Schmidt et al. (2024) indicated that while PXRF technology initially tended to overestimate As concentrations and underestimate Pb concentrations in soil samples compared to ICP-MS analysis, accuracy significantly improved upon applying a ratio correction factor. Even before correction, there was already a strong positive correlation between the calibrated PXRF values and ICP-MS data for both As and Pb, with Spearman coefficients of 0.850 and 0.981,

respectively; however, the use of the correction factor notably enhanced the precision and reliability of the PXRF measurements.

Elements with atomic masses (e.g., Pb) show higher accuracy, while those with smaller atomic masses (e.g., Ni) are expected to exhibit lower intensity given that the XRF yield scales with the atomic number Z, as confirmed by a study by Romzaykina et al. (2024). Lead, Cu, Zn, and Cd were considered reliable for PXRF readings above certain concentration thresholds, while

Ni showed reliability only at higher concentrations. Portable XRF readings generally aligned well with ICP-OES measurements, with slight overestimation observed. Discrepancies, however, were noted for some elements, particularly Cd, further indicating limitations in PXRF accuracy for certain contaminants. Overall, the reviewed studies suggest that PXRF analysis can provide a reliable and efficient alternative to laboratory analysis for heavy metal(loid) contamination assessment in urban soils, with high correlation coefficients reported for varying elements when appropriate correction factors are applied.

Specific values, however, may vary depending on the soil characteristics and analytical protocols used, highlighting the importance of careful calibration instruments for accurate results.

### 3.2 *In Situ* vs. *Ex Situ*: accuracy and precision of PXRF measurements

Portable XRF instruments, widely utilized for rapidly assessing heavy metal contamination in soils globally, allow both *in situ* (field-based) and *ex situ* (laboratory-based) analyses. Popular instrument brands such as Olympus, Thermo-Scientific, Oxford, and Bruker were commonly used in urban soil research, with Olympus and Thermo-Scientific being most frequently reported (Figure 5). While practical factors, including availability and cost, often guide instrument selection, evidence suggests that the specific instrument brand or its age alone has minimal impact on measurement accuracy or precision. Instead, PXRF

performance primarily hinges upon adequate calibration procedures, optimal testing conditions, and appropriate sample preparation methods.

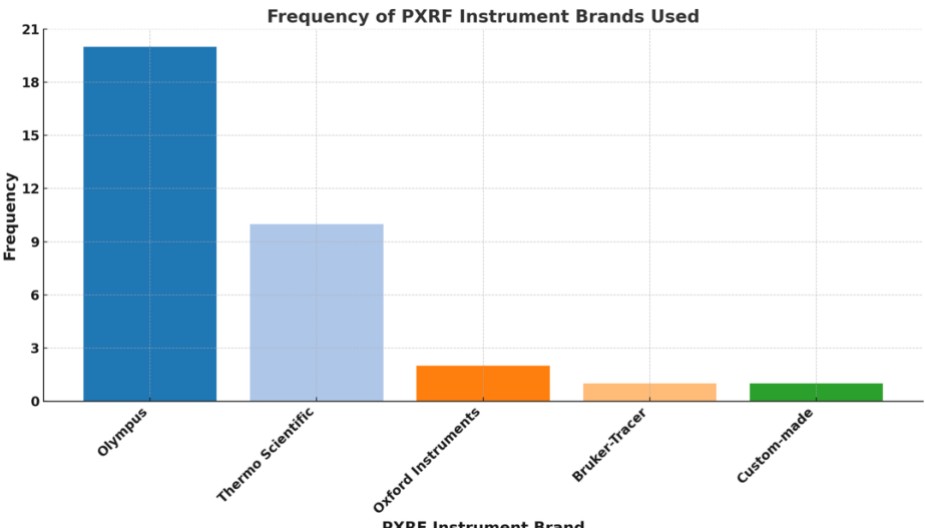

**Figure 5: Frequency of PXRF instrument brands used across reviewed studies. A total of 34 instances noted a specific**
**instrument brand. Only studies that explicitly stated the PXRF instrument used were included in this count.**

Measurement settings have a major impact on PXRF accuracy and precision. *In situ* measurements are fast and convenient, requiring little to no sample preparation, but their accuracy is often compromised by environmental variables like moisture, organic content, and soil heterogeneity. *Ex situ* analyses, by contrast, are conducted in controlled conditions and typically yield
more precise and reliable results. Schumacher et al. (1990) underscored the need for homogenous samples to reduce errors from sample heterogeneity. Hu et al. (2014) showed that in situ PXRF measurements were less accurate and precise compared

to ex situ, yet still reliable when compared with ICP-MS and atomic fluorescence spectrometry (AFS) results. For example, As measurements using in situ and ex situ PXRF were 13% and 31% higher than AFS results, while Pb and Zn values were 38% and 17% lower than ICP-MS values. Copper results showed a 10% lower reading with in situ PXRF and a 15% higher reading with ex situ PXRF compared to ICP-MS, with accuracy generally improving from in situ to ex situ (Hu et al., 2014). Tian et al. (2018) reported, however, poor agreement between in situ and ex situ PXRF measurements compared to ICP-MS after acid digestion (Table 3). Romzaykina et al. (2024) found high $R^2$ values for Pb (0.94), Cu (0.95), and Zn (0.95) with PXRF, which improved with calibration based on ICP-OES. Nickel showed lower accuracy ($R^2 = 0.68$), indicating element-specific reliability. These findings highlight the importance of selecting the appropriate measurement setting based on the intended use. While in situ PXRF can be effective for rapid screening, especially in urban environments where fast decisions are critical, it should not be relied upon for precise quantification without calibration or lab confirmation—particularly for elements like Ni, As, or under variable soil conditions.

Despite these differences, PXRF can yield high accuracy and precision under optimal conditions. Table 3 synthesizes findings from comparative studies, summarizing the overall reliability of PXRF for key elements and highlighting the best and worst conditions for measurement. Lead, for example, is considered "very reliable" when samples are dry, sieved, and measured in ex situ conditions using matrix-matched calibration. Conversely, elements like Hg and Cd exhibit poor reliability under most conditions due to detection challenges and interference from moisture and organic matter. Overall reliability ratings were determined using reported correlation coefficients ($R^2$ values) with traditional laboratory methods and qualitative evaluations of sample conditions. Best measurement conditions generally include dry, homogeneous, fine-grained soils and matrix-matched calibration, while worst conditions involve wet soils, coarse particle sizes, and organic-rich matrices, all of which degrade PXRF performance.

**Table 3: Summary of PXRF measurement reliability for major elements based on comparative study analysis.**

| Element | Overall Reliability | Best Measurement Conditions | Worst Measurement Conditions |
|---------|--------------------|-----------------------------|------------------------------|
| Pb | Very good | Dry, fine-grained, sieved soils; matrix-matched calibration; ex situ or sieved measurements | Wet soils, unsieved in situ measurements |
| Zn | Good | Dry, sieved samples; fine particle size; calibrated standards | Moist soils; organic matter interference |
| Cu | Moderate to good | Dry, calibrated samples; ex situ or sieved preparation | Wet soils; matrix variability |
| As | Moderate | Dry samples with proper calibration | Moisture, organic-rich soils; in situ unsieved |
| Ni | Moderate to good | Dry, sieved samples; standardized calibration | Wet or organic soils |
| Mn | Moderate | Dry, homogeneous, fine-grained soils | Wet samples; coarse particle size |

| Cr | Moderate | Matrix-matched calibration, dry soils | Coarse soils, moisture effects |
| --- | --- | --- | --- |
| **Co** | Moderate | Fine particle size; matrix-matched calibration | Wet soils; organic interference |
| **Cd** | Poor | High concentrations | Very sensitive to matrix/moisture; high variability |
| **Hg** | Very poor | High concentrations | Very high measurement error under all conditions |

To investigate the impact of sample preparation methods on measurements, Gutierrez-Gines et al. (2013) tested soil samples from landfills and abandoned mines. The samples were analyzed using PXRF after two different preparation methods: drying and further grinding and pressing (referred to as "pressed" soil samples). Analyzing fresh samples closely resembled *in situ* determinations, allowing for comparison with prepared samples. Grinding and pressing represented the standard laboratory preparation method for this system, while drying and sieving served as an intermediate step. Comparing measurements on fresh samples with those on dried samples showed high comparability. Notably, the soils in these sites, characterized by a Mediterranean climate, were relatively dry during the sampling period (late spring), with a maximum soil moisture content of about 10%. Given the limited number of urban studies examining both field and lab PXRF measurements across diverse urban environments, future research should focus on refining standardized protocols for sample handling, moisture correction, and calibration.

### 3.3 Advantages and applications of PXRF in urban soil analysis: efficiency, accuracy, and spatial interpolation techniques

The innovative use of PXRF has significantly advanced the field of soil analysis, offering new insights and efficiencies in environmental studies. According to Chakraborty et al. (2017), the portability, inexpensiveness, and accuracy of the PXRF spectrometer offer formidable advantages over traditional laboratory based chemical analyses. In their research, 131 points were scanned and beyond identifying those levels, the optimized spatial variability interpolations were plotted using spherical and exponential kriging models. These interpolations, laid over contemporary high-resolution imagery, allowed him to quickly delineate exactly which areas across the city of Baia Mare exceeded limits for elemental concentrations. Portable XRF reported values clearly identified the pollution hotspots which needed further attention. The primary benefit of this approach is that all described procedures can be conducted on-site within one to two days, allowing for the generation and mapping of results using only a computer (Chakraborty et al., 2017). Finally, many of the inter-elemental correlations established in this research further corroborate the findings of two earlier studies which used PXRF for elemental characterization in Romania (Paulette et al., 2015). In summary, PXRF analysis coupled with spatial visualization of interpolations provide a straightforward approach for delineating soils that are a hazard to the public.

In Romzaykina et al. (2023) study of potentially toxic metals in Moscow's greenspaces, the use of correction factors significantly improved the reliability of contamination maps interpolated from point measurements. These maps were

evaluated by comparing them to reference maps based on ICP-OES data. The application of correction factors reduced the areas with significant deviations (>30%) on the potentially toxic metal maps and increased the fraction of non-deviated areas (±10% from ICP-OES values) by 2 to 6 times, depending on the metal and location. For example, on the Cu map of the RUDN

University campus, areas with significant deviations dropped from nearly 40% on the PXRF-based map to less than 5% on the PXRF × k-based map, as shown in Figure 6. These deviations were linked to different vegetation patches or soil properties, with hotspots in areas with high soil content that interfered with Cu detection by PXRF. Non-deviated areas for Cu, Pb, and Ni increased to 50-70% on adjusted maps (Romzaykina et al., 2023).

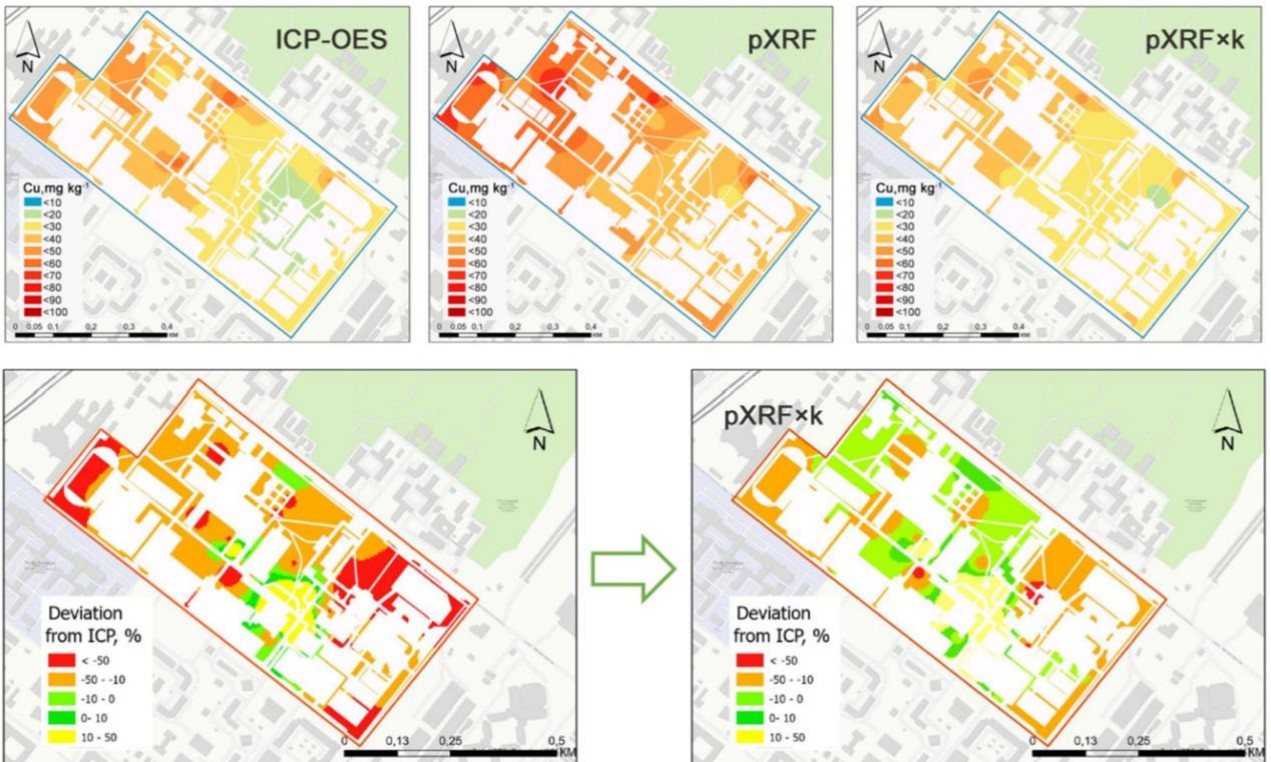

**Figure 6: Maps created by Romzaykina et al. (2024) that show the spatial distribution of deviations in proximal assessments of Cu concentrations from the ICP-OES reference, comparing pXRF without correction factors (left) and with correction factors (right).**

    **Portable XRF analysis can provide accurate and precise results when used appropriately, as demonstrated by the**

**strong correlation observed in several studies. This suggests that PXRF analysis can be a reliable and cost-effective alternative to traditional laboratory methods for measuring heavy metal contamination in urban soils. While ICP and AAS are highly accurate analytical techniques that can detect minute levels of trace substances within a sample, the techniques are complicated, must be carried out by specialists, and the delicate equipment requires regular calibration and maintenance. Alternatively, due to the PXRF's non-destructive nature, samples tested can be reused, preserved,**

**or safely shipped off for further analysis using other methods. In many cases, PXRF analysis is a viable alternative to either replace the aforementioned equipment completely, or act as a backup when it is out of action (Hu et al., 2017).**

Portable XRF offers significant cost advantages over traditional laboratory methods (e.g., ICP-MS, ICP-OES, AAS) for urban soil research, particularly for large-scale or rapid screening applications. Table 5 is a cost comparison based on the reviewed literature.


**Table 5. Cost comparison of using PXRF vs traditional analytical methods based on the reviewed literature.**

| Factor | Portable XRF analyzer | Traditional Methods (ICP-MS/AAS) |
| --- | --- | --- |
| Initial Equipment Cost | $20,000–$40,000 for handheld units | $100,000+ for ICP-MS systems, plus lab infrastructure |
| Per-Sample Cost | $1–$5 (no consumables, minimal prep) | $50–$200 (reagents, acids, lab fees, waste disposal) |
| Labor Time | Minutes per sample (in situ/ex situ) | Hours per sample (digestion, transport, lab analysis) |
| Sample Preparation | Minimal (sieving/drying for ex situ; none for in situ) | Extensive (acid digestion, filtration, hazardous waste management) |
| Throughput | 100–200 samples/day | 10–20 samples/day (limited by lab capacity) |
| Operational Costs | Low (no recurring costs for reagents or waste disposal) | High (chemicals, gases, lab maintenance, waste treatment) |
| Regulatory Compliance | Suitable for screening; lab confirmation needed for definitive results | Required for trace-level analysis and regulatory reporting |

**3.4 Limitations of PXRF analysis: soil heterogeneity, moisture, and organic matter interference**

Factors affecting discrepancies between PXRF measurements of soil standards and their certified values are largely attributed to sample heterogeneity and soil matrix interference, especially during *in situ* testing. For instance, organic materials generally led to overestimation of potentially toxic metals in one study, while mineral substrates yielded more accurate results, with sand being the most accurate substrate. The researchers emphasized the influence of soil matrix on PXRF accuracy, reinforcing the need for calibration in urban environments (Romzaykina et al., 2024). Burlakovs et al. (2015) identified moisture content and

organic matter as factors influencing element concentrations in waste samples, and moisture correction was applied to PXRF raw data to mitigate these effects. In another study, results demonstrated that PXRF measurements are affected by the presence of water and organic matter (Ravansari and Lemke, 2018). For example, it was discovered that increasing water content resulted in decreased recorded concentrations for all elements. This finding highlights the importance of considering soil

moisture effects. Soil moisture can cause errors in PXRF analysis due to increased absorption and scattering (Parsons et al., 2013). The US Environmental Protection Agency reported that soil samples with more than 15% moisture content led to elemental measurements that are lower than the actual concentration (Simmons, 2023).

A study by Rosin et al. (2022) used a control group as well as a manipulated one to investigate the specific effects of water and organic matter content on the PXRF measurements. It was concluded that since the attenuation of X-rays by water is higher than that of air due to greater density of the latter, greater sample moisture leads to lower net peak areas of characteristic X-rays that constitute the sample, which results in lower precision, accuracy, and detection limits. The presence of moisture particularly impacts the accuracy of detecting elements with atomic numbers below 30, including Mg, Al, and Si (Rosin et al., 2022). This study also determined that as the amount of organic matter in a soil sample decreases, the precision and detection limit of the PXRF increases.

Moreover, results demonstrate that the PXRF measurement response is elementally dependent. Ravansari and Lemke (2018) wrote that although the experiments in their study were carefully controlled, they were conducted with a limited number of samples using a single instrument - a Niton XL3t + 950 PXRF in soil analysis mode. Therefore, results should be considered preliminary until they can be verified with a larger number of soils containing a wider range of organic matter fractions, ideally with PXRF analyzers from additional manufacturers (Ravansari and Lemke, 2018).

The presence of other elements in the soil can interfere with the accuracy of PXRF analysis, leading to inaccurate readings. Due to this background noise caused by the soil matrix, Nawar et al. (2019) pointed out that the PXRF can be improved through the spectral data analysis coupled with random forest machine learning method for low-Z elements which have spectral overlap and low fluorescence yield at low concentration (K, P, Ca, and Mg). Furthermore, regular calibration and maintenance of PXRF analyzers is necessary to ensure accurate and precise measurements. Drift in instrument performance over time can lead to errors in readings (Brand and Brand, 2014).

The particle size of the soil can also affect the accuracy of PXRF measurements, which can prove as a barrier in the field. Additionally, PXRF analyzers can only measure heavy metal concentrations in the top few millimeters of soil, further limiting their usefulness in the field at locations that require measurements at greater depths. Heavy metal(loid) contamination in urban soils is often heterogeneous, with hotspots of contamination occurring at specific locations. Portable XRF measurements may not capture the full extent of heavy metal(loid) contamination in such cases.

**3.5 Recommendations and practical considerations when applying PXRF to urban soil research**

Portable XRF analyzers offer significant advantages for urban soil research, including rapid on-site results, cost efficiency, and the ability to screen large areas quickly for heavy metal(loid) contamination. Their utility as a reliable screening tool is

recognized by regulatory agencies and validated by numerous peer-reviewed studies. For instance, EPA Method 6200 indicates that PXRF can reliably identify contamination hotspots with a correlation coefficient (r) of at least 0.7 when compared to confirmatory laboratory methods (ICP-MS, ICP-AES, AAS), affirming its effectiveness for initial field assessments (U.S. EPA, 2007). Typical deviations between PXRF and laboratory analyses are generally within ±20–30%, which is considered acceptable for rapid, field-based evaluations (Palmer et al., 2021; Fedeli et al., 2024). Such performance underscores PXRF's valuable role in preliminary assessments and highlights its potential to reduce laboratory testing needs and costs significantly, especially in urban agricultural settings.(USDA NRCS, 2023)

Calibration checks against certified reference materials generally consider ±20% deviation acceptable for reliable instrument performance (EPA Region 4, 2017; Waikato Regional Council, 2016). Deviations exceeding these thresholds typically result from soil characteristics such as high moisture content, organic matter interference, particle size heterogeneity, and matrix complexity, particularly affecting elements near the PXRF detection limit or those with lighter atomic weights (Parsons et al., 2013; Ravansari & Lemke, 2018). Studies have shown that while PXRF can achieve accuracy within ±20% under controlled laboratory conditions, field conditions often introduce variability, making ±30% deviation more realistic for typical urban soil screening (Fedeli et al., 2024; Shefsky, 1997). Practically, substantial deviations (e.g., 2–3 times difference) pose the risk of inaccurate identification of contamination hotspots.

Furthermore, it is difficult to determine which homogenization method produced the most accurate results, as it can vary depending on the specific samples and the elements being analyzed. In a study by Hu et al. (2014), however, it was observed that grinding and sieving samples before analysis did improve accuracy and precision of results, though for Tian et al. (2018), it only significantly improved accuracy of results for Cu, Mn, and Zn. From examination of results, it can be assumed that there isn't a notable difference between results obtained from grinding soil with a mortar and pestle vs. any form of mechanical grinding.

The accuracy and precision of PXRF measurements can be influenced by several factors, including drying methods, measurement times, and metal(loid) concentrations. Schneider et al. (2016) found that varying measurement times at intervals of 60, 90, 120, 180, and 240 seconds had no significant effect on PXRF accuracy or precision. Tests conducted on four reference materials showed consistent results across different time intervals, with a Friedman test revealing no significant differences in concentrations of 11 elements across the five measurement times at a 5% significance level. Based on these findings, a count time of 60 seconds was deemed sufficient for most analyses.

The type and concentration of metal(loid)s being measured, however, can affect the required testing times. Gutierrez-Gines et al. (2013) demonstrated that longer measurement times improve PXRF accuracy and precision, especially for low metal concentrations. While high metal concentrations produced consistent results even with shorter times, low concentrations

required extended measurement times to enhance detection limits. Additionally, their study highlighted that higher metal concentrations in processed soil samples led to greater accuracy and precision, particularly when longer measurement times were applied.

On the other hand, sample drying methods also play a role in PXRF accuracy. Paulette et al. (2015) found that oven-dried samples yielded moderately more accurate results compared to air-dried samples, possibly due to insufficient drying time for air-dried soils. This suggests that proper sample preparation is essential to improving the reliability of PXRF results.

Optimal analytical conditions for PXRF analysis involve several key factors. Al Maliki et al. (2017) recommended a sample layer thickness of 2 mm, use of special containers (e.g., 6.4 cm Chemplex containers) with plastic films (e.g., 3.6 mm Mylar Polyester), and a moisture content of 0.5% in both the samples and standards. Ensuring a uniform grain size is also crucial for accuracy. The study also noted that various types of sample containers, such as Ziploc bags, can be used effectively as long as consistency is maintained between the standards and the samples. Future research should focus on optimizing the balance between sample bulk density, mineralogy, moisture content, and instrument settings to further enhance PXRF performance.

Therefore, PXRF data should be interpreted with caution. It is recommended that PXRF be employed primarily as a rapid preliminary screening tool to identify potentially contaminated areas, with confirmatory ICP-based laboratory analysis essential for samples critical to decision-making processes (Palmer et al., 2021; U.S. EPA, 2007). This dual approach ensures robust decision-making while benefiting from PXRF's cost-effective and rapid analysis capabilities.

**4 Conclusions**

This review examined the accuracy, precision, and practical utility of PXRF for assessing metal contamination in urban soils, highlighting key factors influencing its effectiveness. Given the increasing role of urban agriculture as a sustainable strategy for enhancing food security, revitalizing urban green spaces, and promoting community resilience, reliable soil contamination assessment is essential. Portable XRF has shown strong capability as a rapid diagnostic tool, particularly for key urban contaminants such as lead, zinc, and copper, when used under optimal analytical conditions—dry, homogenized soil samples, ex situ measurement, and careful calibration. However, performance significantly decreases in field settings characterized by soil moisture, organic content, heterogeneity, and lower contaminant concentrations.

The findings clearly indicate PXRF's suitability as a preliminary screening method within integrated soil management programs, effectively guiding more targeted and cost-effective laboratory analyses. In urban agricultural contexts, this approach allows growers, urban planners, and extension specialists to rapidly identify areas of concern, prioritize soil remediation efforts, and make informed decisions to enhance soil health and crop safety. The USDA-NRCS provides a concise protocol emphasizing clear project objectives, thorough site history evaluation, strategic sampling methods (zigzag, random,

transect), appropriate sample depths aligned with crop roots, and consideration of soil moisture and organic matter. Following these guidelines enhances PXRF accuracy, effectively identifying contamination hotspots needing further detailed analysis or conservation planning (USDA NRCS, 2023).

Several research gaps remain critical for optimizing PXRF use. First, standardized PXRF protocols specifically tailored to urban agricultural soils are lacking, resulting in inconsistent data quality and comparability. Development of comprehensive sampling guidelines and calibration procedures suited to urban conditions would enhance reliability. Second, locally validated correction factors accounting for site-specific soil characteristics—such as variability in organic matter and mineral composition—are insufficiently developed. Additionally, systematic comparisons of PXRF performance across diverse urban
land uses and varying soil depths are scarce, limiting broader applicability.

Future research should address these gaps by developing robust, practical guidelines and region-specific calibration models to improve PXRF accuracy and applicability. Integrating PXRF data with advanced analytical and geospatial approaches—such as remote sensing, spectroscopy, and predictive modeling—could significantly advance soil contamination mapping and urban land management practices. Ultimately, strengthening PXRF methodologies and supporting extension efforts through
improved, accessible soil-testing tools, facilitating healthier urban soils, safer urban food production, and enhanced community well-being.

**Competing interests**

The contact author has declared that none of the authors has any competing interests.

**Acknowledgments**

The authors gratefully acknowledge Dr. Michalis Charilaou for his valuable insights, assistance with calculations, and thoughtful suggestions that significantly improved this manuscript. This study was funded by USDA Cooperative Agreement award number NR223A750025C008.

**Declaration of generative AI and AI-assisted technologies in the writing process**

During the preparation of this work the authors used ChatGPT to improve readability and language of the manuscript. After
550 using this service, the authors reviewed and edited the content as needed and take full responsibility for the content of the published article.

## Author contributions

Eriell Jenkins: Data curation, Formal analysis, Visualization, Writing – original draft; Anna Paltseva: Conceptualization, Funding acquisition, Project administration, Resources, Supervision, Writing – review and editing; John Galbraith: Writing – review and editing.

## Data statement

Data was taken from various sources, as indicated in the description. No original research data was presented.

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
