# Peer review of "Portable X-Ray Fluorescence as a Tool for Urban Soil Contamination Analysis: Accuracy, Precision, and Practicality"

_EGUsphere, 2024_

## Referee Comment (RC2)

Other comments,

*L. 29 I think a "d" is missing in "foo security"
*L. 80 "Articles that provided background information on the PXRF and heavy metal pollution l. 82 was also used in this review." I think it's were instead of was.

* L.82 "During the search, there were articles that appeared via search engine - particularly on Google Scholar - that produced a number of articles that did not meet the criteria set and therefore was not relevant to the stud" sentence not clear
* Figure 1 : not sure about the relevance of 1 flow chart for Wos and 1 for google scholar when the words searched are the same. The horizontal line between urban soil and HM in the WOS flow chart is not horizontal. Please complete the lines or the legend by "and" or "or" or any other logical link they represent.

*Fig. 2 Not sure about the relevance of Fig.2. Maybe a chart with number of studies employing each type of XRF or at least each type of XRF with different specificity as described in paragraph 2.2 would be better?
*L. 262 "Researchers concluded that while XRF measurements can be reliable for certain elements like Pb, Ni, Zn, and Cu, they may not be as accurate for elements like Hg, Cd, Cr, and As, " same sentence than l. 258 ;

* I'm a bit confused about the conclusion for table 1. You mentioned that Ni measurements with ICP and XRF are in close proximity while there is a factor 3 and no R2 ; same for Pb and Zn with a factor 2 between ICP and XRF.

* L. 280 "with Cd showing a slightly increasing trend at higher concentrations" Isn't that Cu rather than Cr?

* L. 294 "Cubist modelling, which helped them obtain predictions of the results. The resulting data exhibited high skewness, with the PXRF having higher values for Lin's Concordance correlation coefficient" please define or explain "cubist modelling" and "lin's concordance"

* L.300 " The results obtained in research conducted by Schmidt et al. (2024) […] PXRF measurements for As and Pb", this look like a list of studies that performed or not … I'm not sure about the relevance of this paragraph after the table and fig.3. Maybe rephrase or complete to highlight the interest of detailing this study? Besides, l.263 the author wrote that method does not perform well for As ….

* Table 2 not sure about "in situ r2" and "ex situ r2" is it the r2 between PXRF and ICP for in -situ (or ex situ) measurements? please detail the legend

* Is paragraph 3.3 a "conclusion" about how to perform measurements ?

 * l. 390 "Portable XRF is effective and economic […]] for each heavy metal individually using traditional laboratory methods." Is that a concluding paragraph ? why an other paragraph of example after this one

* L.430 and 432 comments about the moisture and organic matter content have been wrote before

---

## Author Comment (AC1)

**RC1**: 'Comment on egusphere-2024-3101', Anonymous Referee #1, 28 Nov 2024  reply
The manuscript addresses the review of an interesting topic, however, it lacks the necessary depth to warrant publication.

- **We thank the reviewer for acknowledging the significance of our review topic. We understand the concern regarding the current depth. To address this, we will enhance the manuscript by adding additional analyses, clarifications, and detailed data summaries (as requested in the subsequent comments), which will substantially deepen the discussion and provide the necessary depth for publication.**

While the title and objectives clearly indicate a focus on urban soils, the manuscript does not provide a sufficient justification for limiting the review exclusively to urban soils. Why is the method under review only applicable to urban soils, and would it also be suitable for agricultural soils? The rationale for restricting the review to XRF studies conducted in urban soils is not clearly explained.

- **We agree that the rationale behind limiting our review specifically to urban soils needs clearer articulation. Urban soils present distinct challenges compared to agricultural or natural soils, primarily due to their high heterogeneity, complex land-use history, and diverse contamination sources (e.g., industrial residues, traffic emissions, urban waste). These factors significantly affect analytical precision and accuracy when using portable XRF. Given that urban soils are frequently repurposed for agriculture, recreation, or residential use, the rapid, cost-effective identification of contamination hotspots provided by PXRF becomes especially relevant in urban contexts. We will clarify and expand upon this rationale explicitly in the introduction.**

The introduction sets up expectations for the reader, but the conclusion essentially restates what can already be found in the individual studies reviewed.

- **To address this issue, the revised manuscript will include clearer synthesis statements highlighting novel insights from comparative analyses of the PXRF versus traditional methods. Specific recommendations and limitations derived from recent studies will be more explicitly articulated to underscore practical insights and actionable guidance. Additionally, we will emphasize newly identified gaps and opportunities for future research, providing original insights into PXRF application in urban environments.**

Therefore, as it is currently presented, the manuscript does not offer significant new insights into the limitations and advantages of the method.

The information in Section 2 could be better organized and presented in a more reader-friendly format, such as tables or graphs, to enhance the clarity and accessibility of the bibliographic review results.

- **We acknowledge the importance of clear data visualization and readability highlighted by the reviewer. To improve Section 2, we will provide tables clearly summarizing**

**frequency, accuracy, reliable concentration range, and optimal conditions for PXRF measurements by element.**

To make the review more engaging and to strengthen its case for publication, I suggest the inclusion of the following:

1. A list of the chemical elements analyzed in the reviewed studies, along with their frequency of occurrence.
2. A systematic presentation of $R^2$ values, along with estimates of the accuracy for each element.
3. The detection limits for each element, as reported in the reviewed studies.

- **We agree with this suggestion and will add the following structured information:**
- **A comprehensive table summarizing the correlation coefficients ($R^2$) clearly comparing PXRF and laboratory methods (ICP-MS, ICP-OES, ICP-AES) across reviewed studies, categorized by element.**
- **A dedicated table outlining reported detection limits (LOD) of PXRF instruments for key elements (Pb, As, Cu, Zn, Ni, etc.) based on data explicitly provided in the reviewed literature.**
- **A summarized analysis of PXRF measurements across different studies, highlighting which elements consistently exhibit reliable measurements and under which conditions accuracy significantly decreases.**

Including these details would make the review more informative and comprehensive, offering additional value to the scientific community.

- **Thank you very much for your thoughtful review. The manuscript will be revised according to your suggestions, providing a more substantial rationale, improved clarity, and comprehensive insights that align closely with the manuscript's objectives.**

---

## Author Comment (AC2)

**RC2**: 'Comment on egusphere-2024-3101', Anonymous Referee #2, 07 Feb 2025   reply
This study is relevant and necessary as it provides insight into the performance of portable X-ray devices. However, the article would benefit from a clearer structure and a well-defined research question to better guide the reader.

- **We appreciate this valuable feedback. To enhance clarity, we will explicitly define the main research question at the end of the Introduction section, clearly stating: _"What is the accuracy, precision, and practical applicability of portable X-ray fluorescence (PXRF) in evaluating metal contamination specifically within urban soils, and under what conditions does its performance vary?"_ We will also reorganize the manuscript structure, clearly linking each subsequent section back to this central research question.**

Currently, the inclusion of numerous individual studies makes it difficult to follow, especially since they are not always synthetized or compared to one another—which should be a key objective of a review.

- **We agree with the reviewer's concern. To resolve this, we will restructure and significantly enhance the manuscript by:**
- **Synthesizing key findings of the reviewed studies into concise summary tables (e.g., comparisons of calibration methods, PXRF accuracy by element, and conditions influencing results).**
- **Clearly highlighting commonalities and discrepancies across the reviewed studies, including explicit comparisons of $R^2$ values, detection limits, and element-specific accuracy and precision.**
- **Providing integrative commentary that explicitly compares methodologies, limitations, and practical implications drawn from across the reviewed literature.**

Additionally, the study lacks detail on the choice of the urban setting, which could have a significant impact on the findings.

- **We acknowledge this critical point. We will clearly state that urban soils present specific analytical challenges due to their heterogeneous nature, history of anthropogenic contamination, and complex land-use patterns. We will expand on how these urban-specific factors distinctly impact PXRF performance compared to other contexts such as agricultural or natural soils, thus emphasizing the significance of the urban soil focus of this review.**

Besides, the authors tends to wrote several times some informations for instance about the soil moisture/OM  or mode choice importance. This make the paper hard to read and without any clear onlusion while such study could have been helpful for choice in measurement technic.

- **We agree with this observation. We will carefully revise the manuscript to eliminate redundant statements regarding soil moisture, organic matter, and PXRF operational**

**modes. We will consolidate this information into clearly organized subsections or dedicated paragraphs that systematically address these factors once, providing explicit recommendations and conclusions for each, thereby significantly improving readability and coherence.**

Finally, I'm not conviced about the "performance status" the authors accepted. In some cases there is a factor 2 to 3 between ICP and PXRF measurement which seems quiet high.

- **We thank the reviewer for highlighting this important concern. We will explicitly address the acceptable performance standards more clearly in the manuscript by:**
- **Providing a clear justification of accepted thresholds for differences between PXRF and laboratory (ICP-based) measurements, referencing established guidelines and best practices from the literature.**
- **Clarifying under which circumstances (elements, concentration ranges, measurement settings) these deviations occur, and explicitly discussing their practical implications and limitations.**
- **Proposing clear guidelines and caveats for users interpreting PXRF data in light of these variations, including recommendations for calibration and confirmation of critical findings through ICP-based methods.**

Other comments,
*L. 29 I think a "d" is missing in "foo security"

- **Thank you for pointing this out. The typo will be corrected from "foo security" to "food security".**

*L. 80 "Articles that provided background information on the PXRF and heavy metal pollution l. 82 was also used in this review." I think it's were instead of was.

- **Thank you for this correction. We will revise the sentence to: "Articles that provided background information on PXRF and heavy metal pollution were also used in this review."**

* L.82 "During the search, there were articles that appeared via search engine - particularly on Google Scholar - that produced a number of articles that did not meet the criteria set and therefore was not relevant to the stud" sentence not clear

- **We will rewrite this sentence for greater clarity as follows: "During the literature search, particularly on Google Scholar, a number of retrieved articles did not meet the predefined inclusion criteria and thus were excluded from the review."**

* Figure 1 : not sure about the relevance of 1 flow chart for Wos and 1 for google scholar when the words searched are the same. The horizontal line between urban soil and HM in the WOS

flow chart is not horizontal. Please complete the lines or the legend by "and" or "or" or any other logical link they represent.

- **We appreciate this suggestion for improving Figure 1. To address your comment, we will:**
- **Consolidate the two separate flowcharts into one combined flowchart clearly distinguishing the steps and numbers of articles identified and included from each database.**
- **Clearly indicate logical operators between search terms within the flowchart.**

*Fig. 2 Not sure about the relevance of Fig.2. Maybe a chart with number of studies employing each type of XRF or at least each type of XRF with different specificity as described in paragraph 2.2 would be better?

- **We appreciate the reviewer's suggestion. To address this, we will replace Figure 2 with a clear bar chart illustrating the** number of reviewed studies employing each PXRF instrument type**, as described in Section 2.2.**

*L. 262 "Researchers concluded that while XRF measurements can be reliable for certain elements like Pb, Ni, Zn, and Cu, they may not be as accurate for elements like Hg, Cd, Cr, and As, " same sentence than l. 258 ;

- **We thank the reviewer for pointing out this redundancy. We will carefully revise and remove duplicate sentences to avoid repetition.**

* I'm a bit confused about the conclusion for table 1. You mentioned that Ni measurements with ICP and XRF are in close proximity while there is a factor 3 and no R2 ; same for Pb and Zn with a factor 2 between ICP and XRF.

- **Thank you for highlighting this. We will clarify our interpretation of Table 1 by explicitly addressing these discrepancies.**

* L. 280 "with Cd showing a slightly increasing trend at higher concentrations" Isn't that Cu rather than Cr?
- **Thank you for identifying this. We will carefully verify and correct this statement.**

* L. 294 "Cubist modelling, which helped them obtain predictions of the results. The resulting data exhibited high skewness, with the PXRF having higher values for Lin's Concordance correlation coefficient" please define or explain "cubist modelling" and "lin's concordance"

- **In the revised manuscript, we will briefly define and explain these terms clearly:**
- **Cubist modelling: A statistical machine-learning method used for predicting continuous numeric outcomes from large datasets by creating predictive rule-based models.**

- **Lin's Concordance correlation coefficient (CCC): A statistical measure that quantifies agreement between two measurement methods, indicating both accuracy and precision simultaneously.**

* L.300 " The results obtained in research conducted by Schmidt et al. (2024) […] PXRF measurements for As and Pb", this look like a list of studies that performed or not … I'm not sure about the relevance of this paragraph after the table and fig.3. Maybe rephrase or complete to highlight the interest of detailing this study? Besides, l.263 the author wrote that method does not perform well for As ….

- **We agree with your concern. This paragraph will be rewritten to clearly highlight the significance of Schmidt et al.'s (2024) findings in the context of the review.**

* Table 2 not sure about "in situ r2" and "ex situ r2" is it the r2 between PXRF and ICP for in -situ (or ex situ) measurements? please detail the legend

- **We will improve the table legend accordingly.**

* Is paragraph 3.3 a "conclusion" about how to perform measurements ?

- **Paragraph 3.3 provides recommendations and practical considerations based on the reviewed studies rather than a formal conclusion. We will relabel it to avoid any confusion.**

* l. 390 "Portable XRF is effective and economic […]] for each heavy metal individually using traditional laboratory methods." Is that a concluding paragraph ? why an other paragraph of example after this one

- **We acknowledge this confusion. We will reorganize these sections clearly to separate our concluding statements from specific illustrative examples.**

* L.430 and 432 comments about the moisture and organic matter content have been wrote before

- **Thank you for highlighting this redundancy. We will remove repetitive references and consolidate our discussions of moisture and organic matter clearly into a single, comprehensive subsection to improve readability and clarity.**

---

## Author Response (AR1)

Dear Editor and Reviewers,

We sincerely thank you for your constructive feedback and the opportunity to revise our manuscript. In response to the reviewer comments, we have made several significant improvements that enhance both the depth and clarity of the paper:

- We increased the number of reviewed studies from 67 to 84, broadening the evidence base and strengthening the conclusions.
- A new summary of PXRF instrument brand usage has been added, highlighting Olympus and Thermo Scientific as the most commonly applied, offering practical insight into tool selection.
- We now provide clearer guidance on calibration practices (e.g., use of CRMs), optimal measurement conditions, and testing time with corresponding new figures to support real-world application.
- The revised version concludes with specific research needs, including the lack of urban-specific calibration protocols and the need for integrated spatial methods in future PXRF research.
- Figures and tables have been reorganized (and even some of them were removed) to better communicate key findings, including a refined literature search flowchart and frequency distribution of PXRF instruments.
- We refined the language throughout and restructured sections for better flow, including a clearly stated research question, improved synthesis of results across studies, and stronger integration of practical implications.

Collectively, these changes significantly improve the manuscript's clarity, structure, and practical value, aligning with the goals of both reviewers. We hope the revised version meets your expectations and offers a meaningful contribution to the field of urban soil research.

Sincerely,

Anna Paltseva

**RC1**: 'Comment on egusphere-2024-3101', Anonymous Referee #1, 28 Nov 2024
The manuscript addresses the review of an interesting topic, however, it lacks the necessary depth to warrant publication.

- **We thank the reviewer for recognizing the significance of our review topic. To address concerns about depth, we have expanded the literature base, incorporating a broader and more detailed comparative analysis. We've enhanced methodological discussions, explicitly clarified PXRF limitations, and improved practical recommendations. Additionally, visual aids have been revised to provide clearer data presentations, significantly deepening the manuscript's analytical rigor and practical utility.**

While the title and objectives clearly indicate a focus on urban soils, the manuscript does not provide a sufficient justification for limiting the review exclusively to urban soils. Why is the method under review only applicable to urban soils, and would it also be suitable for agricultural soils? The rationale for restricting the review to XRF studies conducted in urban soils is not clearly explained.

- **We agree with the reviewer on the importance of clearly articulating our rationale. In the revised manuscript, we explicitly emphasize the unique challenges associated with urban soils, such as significant heterogeneity, complex land-use histories, and contamination from diverse anthropogenic sources like industrial residues, traffic emissions, and urban waste. These factors particularly influence PXRF accuracy and precision, making it distinctly suitable for rapid assessment in urban contexts. We now clearly explain that while PXRF can indeed be used in agricultural soils, the specific focus and value of our review lie in addressing these distinctive urban characteristics, thereby providing targeted insights for urban soil management.**

The introduction sets up expectations for the reader, but the conclusion essentially restates what can already be found in the individual studies reviewed.

- **In the updated manuscript, we restructured the conclusions to clearly synthesize comparative insights drawn across reviewed studies, moving beyond mere restatements. We specifically highlight novel insights into optimal methodological approaches, practical limitations, and effective strategies for deploying PXRF in urban soil contexts. Additionally, we've explicitly identified existing research gaps—such as the need for region-specific calibration models and standardized methodological protocols tailored to urban soils—thus offering original guidance for future research and practical applications.**

Therefore, as it is currently presented, the manuscript does not offer significant new insights into the limitations and advantages of the method.

The information in Section 2 could be better organized and presented in a more reader-friendly format, such as tables or graphs, to enhance the clarity and accessibility of the bibliographic review results.

**We appreciate the reviewer's suggestion and have substantially revised Section 2 to improve organization, clarity, and data accessibility. In the updated version:**

- **We have updated a literature search flowchart (Figure 1), now including 84 reviewed studies.**
- **We have replaced the prior citation-based figure with a bar graph (Figure 5) showing the frequency of PXRF instrument brands across studies in section 3.1.**
- **To enhance accessibility of the review data, we included structured tables summarizing analytical parameters by element.**

To make the review more engaging and to strengthen its case for publication, I suggest the inclusion of the following:

1. A list of the chemical elements analyzed in the reviewed studies, along with their frequency of occurrence.
2. A systematic presentation of $R^2$ values, along with estimates of the accuracy for each element.
3. The detection limits for each element, as reported in the reviewed studies.

- **We agree with this suggestion a comprehensive table summarizing the correlation coefficients ($R^2$) clearly comparing PXRF and laboratory methods (ICP-MS, ICP-OES, ICP-AES) across reviewed studies, categorized by element. We also have added a summarized analysis of PXRF measurements across different studies, highlighting which elements consistently exhibit reliable measurements and under which conditions accuracy significantly decreases (Table 2 and 3).**

Including these details would make the review more informative and comprehensive, offering additional value to the scientific community.

- **Thank you very much for your thoughtful review. The manuscript has been revised according to your suggestions, providing a more substantial rationale, improved clarity, and comprehensive insights that align closely with the manuscript's objectives and research question.**

**RC2**: 'Comment on egusphere-2024-3101', Anonymous Referee #2, 07 Feb 2025
This study is relevant and necessary as it provides insight into the performance of portable X-ray devices. However, the article would benefit from a clearer structure and a well-defined research question to better guide the reader.

- **We appreciate this valuable feedback. To enhance clarity, we have defined the main research question at the end of the Introduction section, clearly stating: *"What is the accuracy, precision, and practical applicability of portable X-ray fluorescence (PXRF) in evaluating metal contamination specifically within urban soils, and under what conditions does its performance vary?"* We have also reorganized the manuscript structure, clearly linking each subsequent section back to this central research question.**

Currently, the inclusion of numerous individual studies makes it difficult to follow, especially since they are not always synthetized or compared to one another—which should be a key objective of a review.

- **We fully agree, and have significantly revised the manuscript to better synthesize and compare findings across studies. Specifically, we have:**

- **Introduced summary tables presenting $R^2$ values, analytical methods across elements and instruments (Table 2).**
- **Highlighted cross-study patterns and discrepancies, including conditions where PXRF yields reliable results versus where it underperforms.**
- **Added interpretive commentary throughout, particularly in Sections 3 to draw out key themes, methodological contrasts, and practical implications.**

Additionally, the study lacks detail on the choice of the urban setting, which could have a significant impact on the findings.
- **We acknowledge this critical point. We have clearly stated that urban soils present specific analytical challenges due to their heterogeneous nature, history of anthropogenic contamination, and complex land-use patterns.**

Besides, the authors tends to wrote several times some informations for instance about the soil moisture/OM or mode choice importance. This make the paper hard to read and without any clear onlusion while such study could have been helpful for choice in measurement technic.

- **We agree with this observation. We have revised the manuscript to the best of our capacity significantly improving its readability and coherence.**

Finally, I'm not convinced about the "performance status" the authors accepted. In some cases there is a factor 2 to 3 between ICP and PXRF measurement which seems quiet high.

**In the revised manuscript, we have addressed this concern by:**

- **Clarifying acceptable thresholds for PXRF deviations based on regulatory and peer-reviewed sources (e.g., U.S. EPA Method 6200, which allows up to ±30% for screening-level comparability).**

- **Explicitly discussing cases where PXRF deviates from ICP results—especially for elements with lower atomic mass, or under challenging field conditions.**
- **Providing detailed Tables 2 showing R² values and methods and Table 3 used to help readers assess which metals and conditions yield reliable PXRF results.**
- **New Figures 2 and 3 illustrate how increasing PXRF measurement time significantly lowers detection limits. As shown in Figure 2, elements like Pb, Hg, Cd, and As exhibit up to an ~80% reduction in detection limits when extending scan time from 15 to 480 seconds. Figure 3 further confirms this trend, demonstrating a Poisson-based inverse square root relationship between measurement time and detection limit across ten metals (Kalnicky & Singhvi, 2001). These results emphasize the critical role of count time in improving PXRF sensitivity and analytical precision.**

Other comments,
*L. 29 I think a "d" is missing in "foo security"

- **The typo was corrected from "foo security" to "food security".**

*L. 80 "Articles that provided background information on the PXRF and heavy metal pollution l. 82 was also used in this review." I think it's were instead of was.

- **The sentence was revised accordingly.**

* L.82 "During the search, there were articles that appeared via search engine - particularly on Google Scholar - that produced a number of articles that did not meet the criteria set and therefore was not relevant to the stud" sentence not clear

- **We have rewritten this sentence for greater clarity as follows: "During the literature search, particularly on Google Scholar, a number of retrieved articles did not meet the predefined inclusion criteria and thus were excluded from the review."**

* Figure 1 : not sure about the relevance of 1 flow chart for Wos and 1 for google scholar when the words searched are the same. The horizontal line between urban soil and HM in the WOS flow chart is not horizontal. Please complete the lines or the legend by "and" or "or" or any other logical link they represent.

- **We appreciate this suggestion for improving Figure 1 and incorporated the changes.**

*Fig. 2 Not sure about the relevance of Fig.2. Maybe a chart with number of studies employing each type of XRF or at least each type of XRF with different specificity as described in paragraph 2.2 would be better?

-   **To address this, we have replaced Figure 2 with a clear bar chart illustrating the number of reviewed studies employing each PXRF instrument type and integrated it in section 3.1. We have eliminated section 2.2 to avoid redundancy and lists.**

*L. 262 "Researchers concluded that while XRF measurements can be reliable for certain elements like Pb, Ni, Zn, and Cu, they may not be as accurate for elements like Hg, Cd, Cr, and As, " same sentence than l. 258 ;

-   **We thank the reviewer for pointing out this redundancy. We have revised the text accordingly.**

* I'm a bit confused about the conclusion for table 1. You mentioned that Ni measurements with ICP and XRF are in close proximity while there is a factor 3 and no R2 ; same for Pb and Zn with a factor 2 between ICP and XRF.

-   **Thank you for highlighting this. The Table 1 was corrected to address the discrepancies.**

* L. 280 "with Cd showing a slightly increasing trend at higher concentrations" Isn't that Cu rather than Cr?
-   **Thank you for identifying this. We have corrected this statement.**

* L. 294 "Cubist modelling, which helped them obtain predictions of the results. The resulting data exhibited high skewness, with the PXRF having higher values for Lin's Concordance correlation coefficient" please define or explain "cubist modelling" and "lin's concordance"

-   **In the revised manuscript, we briefly defined and explained these terms:**
-   **Cubist modelling: A statistical machine-learning method used for predicting continuous numeric outcomes from large datasets by creating predictive rule-based models.**
-   **Lin's Concordance correlation coefficient (CCC): A statistical measure that quantifies agreement between two measurement methods, indicating both accuracy and precision simultaneously.**

* L.300 " The results obtained in research conducted by Schmidt et al. (2024) […] PXRF measurements for As and Pb", this look like a list of studies that performed or not … I'm not sure about the relevance of this paragraph after the table and fig.3. Maybe rephrase or complete to highlight the interest of detailing this study? Besides, l.263 the author wrote that method does not perform well for As ….

-   **We agree with your concern. This paragraph was rewritten.**

* Table 2 not sure about "in situ r2" and "ex situ r2" is it the r2 between PXRF and ICP for in -situ (or ex situ) measurements? please detail the legend

-   **We have removed Table 2 for the manuscript.**

* Is paragraph 3.3 a "conclusion" about how to perform measurements ?

- **Paragraph 3.3 was moved to 3.5 section and provides recommendations and practical considerations based on the reviewed studies rather than a formal conclusion.**

* l. 390 "Portable XRF is effective and economic […]] for each heavy metal individually using traditional laboratory methods." Is that a concluding paragraph ? why an other paragraph of example after this one

- **We acknowledge this confusion and reorganized these sections clearly to separate our concluding statements from specific illustrative examples.**

* L.430 and 432 comments about the moisture and organic matter content have been wrote before

- **Thank you for highlighting this redundancy. We have made changes in the text to avoid redundancy.**